# Optimization and Rheological Study of an Exopolysaccharide Obtained from Fermented Mature Coconut Water with *Lipomyces starkeyi*

**DOI:** 10.3390/foods11070999

**Published:** 2022-03-29

**Authors:** Yilin Guo, Wentian Li, Haiming Chen, Weijun Chen, Ming Zhang, Qiuping Zhong, Wenxue Chen

**Affiliations:** 1College of Food Sciences & Engineering, Hainan University, 58 People Road, Haikou 570228, China; 20190881310118@hainanu.edu.cn (Y.G.); 19085231210019@hainanu.edu.cn (W.L.); chenwj@hainanu.edu.cn (W.C.); 995065@hainanu.edu.cn (M.Z.); 990511@hainanu.edu.cn (Q.Z.); chwx@hainanu.edu.cn (W.C.); 2Maritime Academy, Hainan Vocational University of Science and Technology, 18 Qiongshan Road, Haikou 571126, China

**Keywords:** mature coconut water, exopolysaccharide, response surface optimization, rheological properties, interface adsorption

## Abstract

The current research aimed to solve the environmental pollution of mature coconut water by *Lipomyces starkeyi* and provide a study of its high value utilization. The innovation firstly investigated the rheological properties and interface behavior of a crude exopolysaccharide and provided a technical support for its application in food. A response surface methodology was performed to ameliorate the fermentation factors of the new exopolysaccharide with mature coconut water as a substrate, and the consequences suggested that the maximum yield was 7.76 g/L under optimal conditions. Rotary shear measurements were used to study the influence of four variables on the viscosity of the exopolysaccharide solution. The results illustrated that the exopolysaccharide solution demonstrated a shear-thinning behavior and satisfactory thermal stability within the test range. The viscosity of the exopolysaccharide solution was significantly affected by ionic strength and pH; it reached the peak viscosity when the concentration of NaCl was 0.1 mol/L and the pH was neutral. The adsorption behavior of the exopolysaccharide at the medium chain triglyceride–water interface was investigated by a quartz crystal microbalance with a dissipation detector. The results demonstrated that the exopolysaccharide might form a multilayer adsorption layer, and the thickness of the adsorption layer was at its maximum at a concentration of 1.0%, while the interfacial film was the most rigid at a concentration of 0.4%. Overall, these results suggest that the exopolysaccharide produced by *Lipomyces starkeyi* is an excellent biomaterial for usage in drink, makeup and drug fabrications as a thickening and stabilizing agent.

## 1. Introduction

Mature coconut water contains more salts and proteins than immature coconut water [1], while the decrease in water content and the increase in crude fat content result in the flavor deterioration of coconut water during the ripening process [2]. Considering that mature coconut water contains organic nutrients, it meets the nutrient requirement and supports the multiplication of microbes [3]. Therefore, mature coconut water is used as an inexpensive nutritional supplement in many media of microbes [4] More specifically, mature coconut water can be utilized to produce nata de coco through the fermentation of *Acetobacter xylinus* at room temperature in some tropical regions [5,6]. However, in many regions (such as Hainan Province, China), a higher cost is required for the growth and fermentation of *Acetobacter xylinus* at room temperature, which limits the production of nata de coco [7]. Therefore, plenty of matured coconut water is usually discarded during the coconut processing, resulting in a massive waste of resources and serious environmental pollution in Hainan Province [8]. From this, exploring an applicable microbe to replace *Acetobacter xylinus* is an urgent matter in order to make full use of mature coconut water.

The lipid-producing microorganism *Lipomyces starkeyi* (*L. starkeyi*) is a strain primordially obtained from the natural environment and belongs to a type of yeast [9]. *L. starkeyi* grows relatively fast and is suitable for growing at 30 °C [10]. In addition, it can use a wide range of carbohydrates for life-metabolic activities [11]. Therefore, it can be used as a potential substitute for *Acetobacter xylinus*. Almost all previous studies of *L. starkeyi* concentrated on its production of microbial oils [11,12,13,14]. Little attention was paid to its high production capacity of polysaccharides. The basic structure of the purified exopolysaccharide produced by *L. starkeyi* was investigated in previous studies [7,9]. However, the yield optimization, rheological properties, and interfacial behavior of the crude exopolysaccharide yield by *L. starkeyi* was not investigated in our previous study. Microbial exopolysaccharides are more interesting than plant polysaccharides because of their efficient production, nontoxicity, biocompatibility, and no limitations regarding climate, regions, and diseases [15]. Therefore, the exopolysaccharides are always used in drink, makeup, and drug fabrications as additives and excipients [15,16]. Rheological properties are important characters, which can be employed to evaluate whether the exopolysaccharide can be a good candidate for usage in cosmetics or food [17,18,19,20].

We hypothesized that the yield of the crude exopolysaccharide could be significantly increased by optimizing fermentation conditions. In addition, we speculated that different physicochemical factors and ion types may create various influences on the apparent viscosity of the crude exopolysaccharide aqueous solution. In addition, we also wanted to investigate the adsorption process and the properties of the formed interface layer at the medium chain triglyceride (MCT)–water interface. In the present investigation, the response surface methodology (RSM) was employed to ameliorate the culture factors of *L. starkeyi,* taking the yield of its crude exopolysaccharide (LSEP) as an index. Furthermore, the rheological properties of crude LSEP were studied, focusing on the influences of heat, pH, NaCl, and CaCl_2_ on the apparent viscosity. The quartz crystal microbalance with a dissipation detector (QCM-D) was employed to investigate the adhesion behavior of crude LSEP at the MCT–water interface. What is innovative about this study is that we firstly study the rheological properties and interface behaviors of crude LSEP, which provide a technical support for its high value application and solve the waste of mature coconut water.

## 2. Materials and Methods

### 2.1. Materials and Chemicals

The experimental strain (*L. starkeyi* 58680) was acquired from the American Type Culture Collection (ATCC, Manassas, VA USA). The composition of the yeast culture medium was peptone (5.0 g/L), malt extract (3.0 g/L), dextrose (10.0 g/L), and yeast extract (3.0 g/L). The fermentation medium was prepared by dissolving glucose (90 g/L), KH_2_PO_4_ (0.35 g/L), Na_2_HPO_4_ (0.125 g/L), (NH)_4_SO_4_ (1.5 g/L), MgSO_4_ (1.5 g/L), CaCl_2_ (0.1 g/L), FeSO_4_·7H_2_O (0.0082 g/L), ZnSO_4_·7H_2_O (0.01 g/L), MnSO_4_·H_2_O (0.01 g/L), and CuSO_4_ (0.01 g/L) in mature coconut water. The reagents used in the study were bought from the Guangzhou Chemical Reagent Factory (Guangzhou, China). Unless otherwise stated, all of the chemicals used were of analytical grade.

### 2.2. Fermentation Conditions

The *L. starkeyi* were preserved at 4 °C in a constant temperature incubator (Tai Hong LRH-250A, Shaoguan China). In order to prepare the seed medium, the strain was transferred from a YM-agar slant (yeast culture medium containing 2% agar) to a sterilized YM medium (125 mL in a conical flask of 500 mL). Then, the seed was incubated in a biological constant temperature culture shaker (30 °C, 140 rpm) for 48 h. After that, the seed was injected in the pre-prepared sterilized mature coconut water, which was subsequently incubated in a biological constant temperature culture shaker (30 °C and 140 rpm) for 6 days. The pre-configured NaOH (0.5 mol/L) was utilized to hold the pH (5.5) throughout the fermentation [9].

### 2.3. Extraction of Crude LSEP

According to a previous study, the crude LSEP was isolated and extracted by the property of being insoluble in ethanol [21]. After 6 days of cultivation, the fermented broth was centrifuged (12,000× *g*, 10 min) to remove the *L. starkeyi* and insoluble impurities. After centrifugation, a moderate amount of trichloroacetic acid was dissolved in the supernatant and then placed at 4 °C for 6 h to denaturalize the protein, which was then removed by centrifugation. For the extraction of crude LSEP, pre-cooled 98% ethanol (4 °C) was mixed with the above centrifuged solution (1:3, *v*/*v*) and placed at 4 °C for about 12–14 h. After alcohol precipitation, the LSEP was lyophilized [7]. Then, the LSEP was dialyzed in ultrapure water with the help of a dialysis tube (Mw cutoff 1000 Da, Shanghai yuanye Bio-Technology Co., Ltd, Shanghai, China. After lyophilizing, the crude LSEP was obtained and stored for later use.

### 2.4. Single-Factor Experiment

There are many conditions that can affect the exopolysaccharide yield of *L. starkeyi*. Four factors (A: concentration of glucose; B: pH; C: concentration of Mn^2+^; D: concentration of Zn^2+^) were selected in this experiment to investigate their effect on the yield of LSEP during fermentation. The mature coconut water was fermented with a designated concentration of glucose (0, 30, 60, 90, and 120 mol/L, respectively), pH (5.5, 6, 6.5, 7, and 7.5, respectively), concentration of Mn^2+^ (5, 10, 15, 20, and 25 mol/L, respectively), and concentration of Zn^2+^ (5, 10, 15, 20, and 25 mol/L, respectively). Each experiment was performed in triplicates.

### 2.5. Optimization of Crude LSEP Production

The response surface methodology (RSM) was used to determine the relationship among the selected experimental factors and expected responses [22]. According to the outcomes of the aforesaid single-factor experiment, a Box-Behnken design (BBD) was performed and analyzed by Design-Expert 12 software (Stat-Ease, Minneapolis, MN, USA). Four factors (A: concentration of glucose, B: pH, C: concentration of Mn^2+^, D: concentration of Zn^2+^) and three-level response surface tests were then carried out to find the optimal values of the variables.

According to the 29 groups of experiments designed by Box-Behnken, quadratic multinomial regression equations were established. In order to establish the regression equation, the test factors were coded according to the equation:*x_i_* = (*x^i^
* − *x_i_^x^*)/∆*x^i^*(1)

The *x_i_* represents the coded value of the *i*th independent variable, *x^i^* represents the real value of the *i*th independent variable, *x_i_^x^* represents its value in the center point of the interval, and ∆*x^i^* represents the step change value:Y = b_0_ + ∑*_i_* b*_i_ x_i_* + ∑*_i_* ∑*_j_* b*_ij_ x_i_x_j_* + ∑*_i_* b*_ii_ x_i_*^2^
(2)

The Y represents the observed response, b_0_ represents an intercept, b*_i_* represents the first-order model coefficient, b*_ii_* represents the quadratic coefficient for the factor *i*, b*_ij_* represents the linear model coefficient for the interaction between factors *i* and *j*. The variable *x_i_x_j_* represents the first-order interactions between *x_i_* and *x_j_
*(*i* < *j*).

Then, the significance of each item in the regression equation was tested, and the analysis of variance was performed by using Design-Expert 12 software (Stat-Ease, Minneapolis, MN, USA). The interaction of each factor was observed through a response surface analysis. In the end, the optimum fermentation condition of *L. starkeyi* was obtained.

### 2.6. Rheological Measurement of Crude LSEP

The rheological properties of the crude LSEP at different pH, NaCl, and CaCl_2_ concentrations were studied by using a rheometer with a 60 mm conical plate (2°) (HAAKE MARS 40, Thermo Fisher, Waltham, MA USA). The aqueous solutions of LSEP (0.8%, *w*/*v*) were prepared and the pH was adjusted to preset values (3, 5, 7, and 9, respectively). The plots of viscosity with the shear rate and the viscosity with temperature were measured. The aqueous solutions of LSEP (0.8%, *w*/*v*) were prepared and different amounts of NaCl or CaCl_2_ were added (0–0.4 mol/L), respectively. When tested, the distance between the rotor and the plate was 1 mm. The appropriate amount (1 mL) of each sample was added during all the tests. The viscosity-shear rate curves were monitored in the shear rate range of 0.1–100 s^−1^ and all of the measurements were tested at 25.0 ± 0.1 °C. The viscosity–temperature curves were measured at a range of 20–90 °C and the heat-up speed was 14 °C/min [23].

### 2.7. Adsorption of Crude LSEP at MCT–Water Interface

The quartz crystal microbalance with the dissipation detector (QCM-D) model (E4) from Q-Sense (Biolin Scientific AB, GOT Sweden) was employed to investigate the adhesion process of crude LSEP at the MCT–water interface. Before the coating of MCT, all sensors were immersed in a mixed liquor made up of ultrapure water, 25% ammonia, and 30% hydrogen peroxide (5:1:1) at 75 °C for 15 min. The MCT was dissolved in chloroform (0.5%, *w*/*v*) and the MCT–chloroform solution (300 μL) was then coated on the surface of a sensor at 2000 rpm for 30 s. Subsequently, the sensors were placed in a vacuum drying oven at 40 °C for 12 h. The QCM-D experiment was conducted at 25 °C with a flow rate of 80 μL/min. After establishing the stable frequency, f, and the dissipation, D, of the baseline, the adsorption of LSEP at the MCT–water interface was determined. After about 160 min, deionized water was used again to wash away the uncombined or incompact LSEP when the adsorption curve was stable [24].

## 3. Results

### 3.1. Single-Factor Experiment

In this study, the effects of glucose concentration, Zn^2+^ concentration, Mn^2+^ concentration, and pH on the production of exopolysaccharide were analyzed by using single-factor experiments. As can be observed in Figure 1a, the production of LSEP improved with the increase in glucose concentration in the range of 0–90 g/L, while the yield of LSEP decreased when the concentration of glucose was above 90 g/L. As the concentrations of Zn^2+^ and Mn^2+^ increased, the yield increased first and then declined when the concentration was higher than 10 mg/L (Figure 1b,c). As shown in Figure 1d, five pH values (5.5, 6.0, 6.5, 7.0, and 7.5) were employed to investigate the influence of pH on the LSEP yield. It was easy to find that the LSEP yield improved significantly with the increase in pH while decreasing sharply when pH exceeded 6.5.

### 3.2. Optimization of Crude LSEP Yield

#### 3.2.1. Analysis of the Model

The process variables and their ranges of the BBD experiment are displayed in Table 1. The BBD experimental design list, composed of 29 runs with the corresponding observed values, is shown in Table 2. By a performing multivariate regression analysis and significance tests on the experimental data, the functional relationship between variables and the response value was as follows: Y = +5.81 +1.53A + 0.4000B + 0.4833C + 0.0500D + 0.2250AB + 0.2125AC + 0.1625AD + 0.0125BC − 0.1625BD + 0.1000CD − 0.2738A^2^ − 0.3738B^2^ − 0.1863C^2^ − 0.5613D^2^.

The ANOVA details of the fitted equation for the production of LSEP are displayed in Table 3. The F value of the model was 22.79, demonstrating that the regression model was extremely significant because of *p* < 0.0001 [25]; namely, the probability caused by noise was very small. The error term results were not significant (*p* > 0.05), demonstrating that the regression equation conformed to the actual situation well and the model could be used to predict the test results.

#### 3.2.2. Analysis of Response Surface

In this study, the 3D plots were obtained by keeping two of the variables fixed while the other two are varying (Figure 2a–f). Figure 2a shows that the maximum yield of LSEP (7.3 g/L) was obtained when the pH was 7.5 and the concentration of glucose was at the maximum (120 g/L). The yield of LSEP exhibited a rising tendency with the increase in glucose concentration (Figure 2b,c). The maximum output of LSEP was obtained at the maximum concentrations of Mn^2^/Zn^2+^ (15 mg/L) and glucose (120 g/L). Figure 2d shows that when the concentration of Mn^2+^ was set, the yield of LSEP improved with the increase in pH (from 5.5 to 6.5). It was worth noting that the yield of LSEP decreased when the pH was beyond 6.5. The contour lines (Figure 2d) were approximately round, indicating that the interaction between Mn^2+^ and pH was not significant and the highest yield reached 5.5 g/L (pH 5.5, 10 mg/L of the Mn^2+^). In Figure 2e, the 3D response surface plot was developed for the yield with varying pH and Zn^2+^. The ellipticity of the contour map was small, indicating that the interaction between pH and Zn^2+^ was weak [26]. As shown in Figure 2f, the yield was only affected by the concentrations of Mn^2+^ and Zn^2+^. When the concentration of Zn^2+^ was constant, the production of LSEP improved gradually with the increasing concentration of Mn^2+^. When the concentration of Mn^2+^ was constant, the production of LSEP improved significantly with the rise of Zn^2+^. However, the yield decreased when the content of Zn^2+^ was beyond 10 mg/L. Through the optimization of the response surface design experiment, the optimal fermentation condition was predicted as follows: 120 g/L of glucose, 14.89 mg/L of Mn^2+^, 11.67 mg/L of Zn^2+^, and pH 6.8.

### 3.3. Rheological Properties of Crude LSEP

The rheological characters of exopolysaccharides are rested with their construction and molecular size, which were influenced by different conditions, such as salts, temperature, and pH [27]. At the same concentration (0.8%) and shear rate, the aqueous solutions of LSEP exhibited different thickening abilities and different apparent viscosities by varying the pH and salt concentrations. As can be observed from Figure 3a,c,e, regardless of different pH or salt concentrations, the viscosity declined with the acceleration of shear rate. The shear-thinning characteristics of LSEP solutions demonstrated typical non-Newtonian fluid properties.

As shown in Figure 3b,d,f, an increase in temperature can accelerate the thermal movement among molecules in the LSEP solution, which increased the intermolecular distance and weakened the interactions, causing the apparent viscosity of the solution to drop slowly [28]. Figure 3a,b shows that the influence of pH on the viscosity of LESP solution was great, compared with the influence of different salt concentrations. The viscosity of LESP solution firstly rose with the increase in pH and then decreased when the solution was alkaline. When the pH was 7, the viscosity reached the maximum. In the process when pH was extremely small, the viscosity of the solution reduced significantly and even a phase separation occurred.

Figure 3c–f described the effect of ionic concentrations on the viscosity of LSEP solutions at various shear rates. The results indicated that the LSEP solution still maintained the property of a non-Newtonian fluid when NaCl and CaCl_2_ were added. In Figure 3c,d, the viscosity of the LESP solution increased when increasing the CaCl_2_ concentration. The rheological characteristics of the LSEP solution can be regulated by the amount of NaCl. As shown in Figure 3e,f, the viscosity of the LESP solution reached the maximum when the concentration of NaCl was 0.1 mol/L.

### 3.4. Adsorption of Crude LSEP at the MCT–Water Interface

QCM-D was used to analyze the adhesion behavior and thickness of the interfacial film at the MCT–water interface. As shown in Figure 4a,b, a positive f and a negative D were caused by the adhesion of LSEP. As a result of the higher adhesion and cohesion of the LSEP solution versus the ultrapure water, the primal f decreased. Subsequently, a slowly linear deposition occurred, which cannot lead to an adsorption equilibrium until 90 min. The f value rose rapidly after rinsing, suggesting that the loosely bound LSEP was washed away by deionized water [24]. It was worth noting that the D values were higher than 1 × 10^6^ (Figure 4b), illustrating that the interfacial film was viscoelastic [29]. The significant drop in D during rinsing indicated that the construction of the reserved interfacial film turned became more rigid and possibly tighter than that before the unbound portion was washed away. Figure 4c shows the thickness of the interfacial film during the adsorption and desorption.

The D-f plots, plotted according to the ratio between D and f (Figure 4d), illustrated the changes in polymer conformation as a qualitative analysis of the interfacial layer. The K_1_, K_2_ and K_3_, K_4_ represent the slopes of the adsorption and rinsing (Table 4), respectively. Furthermore, the dynamic change processes of interfacial adsorption can similarly be observed in D-f plots. As can be observed, the adsorption process (0.4%) was almost linear, meaning no conformational change occurred. Beyond that, all the concentrations demonstrated quick adhesions first and then demonstrated tardy adhesion after the turning point, which was a sign of possible multilayer adsorption. The lowest slope of K_1_ reached the lowest concentration (0.4%), suggesting that interfacial film was the most rigid. Compared to the adsorption process, all slopes of the D-f curves changed significantly in the rinsing process.

## 4. Discussion

As is well-known, carbon and its concentration usually play an important role because the nutrient is directly involved in cell proliferation and metabolite biosynthesis [30]. The yield of crude LSEP improved with the increase in glucose concentration, which may because the glucose provided enough carbon source for the *L. starkeyi*, resulting in the increase in cell number and then increasing the yield of LSEP. When the level of glucose was above 90 g/L, the yield of crude LSEP declined because the adequate supply of carbon sources caused the *L. starkeyi* to produce a large amount of lipid, which cut down the production of LSEP [9]. The increase in crude LSEP yield may be because Zn^2+^ and Mn^2+^ are indispensable to the proliferation and metabolic activity of *L. starkeyi* [14] and can increase the rate of cell division [31]. The sufficiency of Zn^2+^ and Mn^2+^ can make the cell proliferate massively, thus the yield of crude LSEP increased with the increase in Zn^2+^ and Mn^2+^. Meanwhile, the lack of Zn^2+^ and Mn^2+^ increased lipid production, which generally seemed to share an inverse relationship with the exopolysaccharide content [9]. It may because excessive quantities of Zn^2+^ and Mn^2+^ were toxic, resulting in decreased growth or increased death [31]. The pH of the medium can also affect cell growth and the production of metabolites, which may be because pH can affect the permeability of the cell membrane and the activity of some enzymes [30].

The pH and the concentrations of glucose, Zn^2+^, Mn^2+^ were further studied with the BBD experiment to illustrate the interactions among them. In general, the lack of fit is a sign of model failure, representing experimental data that were not included in the regression, or variables that cannot be explained as random errors [32]. In this study, the lack of fit was 0.1356, indicating that the derived second-order equation can be used to deduce the experimental combinations of the crude LSEP yield. The reliability analysis of the regression model (R^2^ = 0.9580) demonstrated that the change (95.80%) in the response value of the model came from independent variables, revealing a high pertinence between the predicted value and the measured value. The corrected correlation factor (Adj-R^2^ = 0.9159) similarly demonstrated a high degree of reliability and fit. The small coefficient of variation (C.V. = 6.45%) demonstrated that the error between theoretical and actual values was small, clearly indicating the high accuracy and great confidence level of the theoretical values. The Adeq accuracy measured the signal-to-noise ratio and a percentage higher than 4 was satisfying. The rate (17.54) indicated a satisfactory signal, which demonstrated that the model was meaningful for the fermentation process. The above results indicated that this derived second-order equation can be employed to speculate the crude LSEP yield [33]. In general, unless the model exhibits a good fit, optimization via response surfaces may give poorer or misleading results, which means that testing the accuracy of the derived second-order equation is indispensable [34]. A deeper analysis of the model demonstrated that the yield of crude LSEP reached an extremely significant level. As shown in the model, the primary terms A, B, and C reached a highly significant degree (*p* < 0.01). That is to say, the pH, concentrations of glucose and Mn^2+^ showed extremely significant effects on the yield. The influence of square terms B^2^ and D^2^ on crude LSEP production reached a significant degree (*p* < 0.05) and extremely significant level (*p* < 0.01), respectively. Furthermore, the influence order of the four elements on the production of crude LSEP was A > C > B > D. The 3D plots of RSM provide a way to visualize prediction models that reflect the relation between observations and various factors [35]. Under the improved conditions, the maximum production of crude LSEP was 7.76 g/L. Compared with the previous reports, the exopolysaccharide yield of *L. starkeyi* was higher than those of *Providencia* sp. (3.92 g/L) and *Phoma dimorpha* (5.48 g/L) [35].

As we all know, the rheological properties of microbial exopolysaccharides in aqueous solutions are very important for their application in industrial production. Therefore, the evaluation of the rheological properties must be conducted [36]. The construction of huge chain molecules, which were composed by macromolecular polymer particles that made the huge chain molecules hook and entangle each other, forming flow resistance between particles and ultimately resulting in a higher apparent viscosity. This was more viscous at a steady state or a low flow rate, but thinned at a higher flow rate. As a result, the solution of LSEP exhibited a typical shear-thinning behavior. Secondly, the interaction between long-chain disordered polymers weakened, which resulted in the decrease in viscosity [37]. Furthermore, some chemical forces (such as hydrogen and electrostatic and hydrophobic force) among molecules weakened at thermal action, which demonstrated an adverse impact on apparent viscosity. That was why the viscosity-temperature curves went down slightly. However, the change of temperature had little influence on the apparent viscosity of crude LESP solution, indicating that the viscosity of crude LESP solution had a good temperature resistance. This property is important for processing applications and has a potential value as a food thickener and stabilizer. The viscosity of the crude LSEP solution was low under acidic or alkaline conditions, which may because they led to hydrogen bond breakage of the LSEP, promoting the disintegration, reducing the molecular weight, and eventually resulting in decreasing the apparent viscosity [37]. An analogous tendency was also observed in the aqueous solution of a kind of polysaccharide from *Tremella foliacea* [38] and polysaccharide from the Auricularia auriculata fungus [39]. The ion concentration is an important parameter to evaluate the functional rheological properties of crude LESP [37]. The viscosity of the crude LESP solution increased with the increase in CaCl_2_ (Figure 3c,d), which might be due to the sufficient cations shielding the negative charges on the chain segments, thereby reducing the electrostatic repulsion between LSEP molecules and facilitating the formation of spiral aggregates. On the other hand, the addition of cations might change the polarity of the aqueous solution, which reduced the solubility of LSEP. The NaCl decreased the charge of polysaccharides and increased the formation of the solvable metal complex, thus resulting in the increase in viscosity [37]. Oliveira et al. [40] found that excess sodium had a negative effect on the viscosity of solutions. Therefore, as the monovalent ion increased, the polymer chain shrank, resulting in a decline in viscosity. This conclusion is similar to the study of Xu et al. [41]. In the presence of salts, the apparent viscosity of the crude LESP solution decreased, demonstrating the typical behavior of a polyelectrolyte. However, ionic force is not the only effect that determines the apparent viscosity of exopolysaccharide aqueous solutions; it is also closely related to the type of inorganic or organic cations. In general, the apparent viscosity is proportional to the valence state of the cation at a similar ionic force. Therefore, the viscosity of the crude LSEP solution varied with the addition of NaCl and CaCl_2_.

The thickness of the interfacial film is a critical indicator in evaluating the stability of the emulsion. Moreover, the interfacial film can explain the emulsification mechanism at a microscopic level [42]. Generally speaking, the thicker the interfacial film, the better the emulsifying performance of the emulsifier. The formation of the interfacial film is a process from loose to dense. With the continuous adsorption of emulsifiers, intermolecular aggregation led to the formation of the microstructure. The repulsive effect of the bound water molecules increased the rigidity of the interface film during the adsorption of surfactant molecules to the interface. The increase in thickness with time, especially when the concentration of crude LSEP was 1.0%, was mainly because of the dense molecular adsorption and rearrangement. The D-f plots can directly reflect the properties of the interfacial layer under different concentrations of crude LSEP than the individual D and f plot [29]. Generally speaking, a higher D/f value indicates more energy loss per frequency unit and the formation of a more swellable interfacial film, while a lower slope value of the D-f plot indicates a relatively thinner but more rigid adsorbed layer. A constant D-f plot slope means no conformational variation happened during adhesion or rinsing. However, a conformational variation happened in the interfacial film if the D-f plot slope is nonlinear or has a discontinuity. Generally speaking, a discrete curve means ‘quick’ adhesion, whereas a consecutive curve means ‘tardy’ adhesion [24]. For the rinsing process, all the slopes of K_3_ were very big before the kinks, mainly because of the reduced adhesion and cohesion of the mobile phase. However, all the slopes of K_4_ became smaller, demonstrating that all the interfacial films become more compact due to the desorption of uncombined or incompact LSEP [43].

## 5. Conclusions

The yield of crude LSEP extracted from *L. starkeyi* was optimized by BBD design and the maximum yield was 7.76 g/L under the optimal condition (120 g/L of glucose, 14.89 mg/L of Mn^2+^, 11.67 mg/L of Zn^2+^, and pH 6.8). The viscosity of the crude LSEP solution exhibited a non-Newtonian flow behavior and good thermal stability. In addition, the effect of pH and the concentrations of NaCl and CaCl_2_ on the apparent viscosity of the crude LESP solution was studied, suggesting that the viscosity reached a peak value at pH 7. In addition, the viscosity of crude LSEP solutions was proportion to the concentration of CaCl_2_ in the range of 0–0.4%, while approaching the peak level when the concentration of NaCl was 0.1%. The results of QCM-D illustrated that crude LSEP can form a viscoelastic multilayer on the MCT–water interface. The lowest slope of K_1_ at the concentration of 0.4% indicated that the interfacial film was the most rigid. This study indicated that crude LESP may have potential application value as a good stabilizer and thickener in the food industry and pharmacy.

## Figures and Tables

**Figure 1 foods-11-00999-f001:**
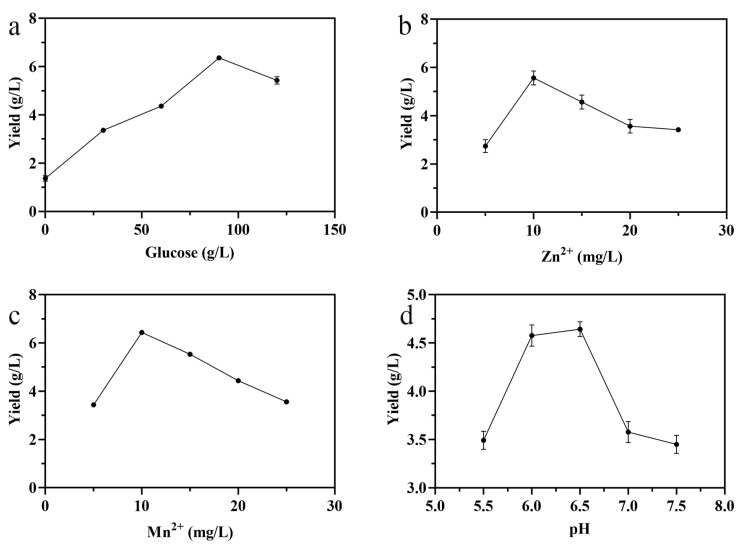
Effects of different concentrations of glucose (**a**), Zn^2+^ (**b**), Mn^2+^ (**c**), and pH (**d**) on the production of LSEP.

**Figure 2 foods-11-00999-f002:**
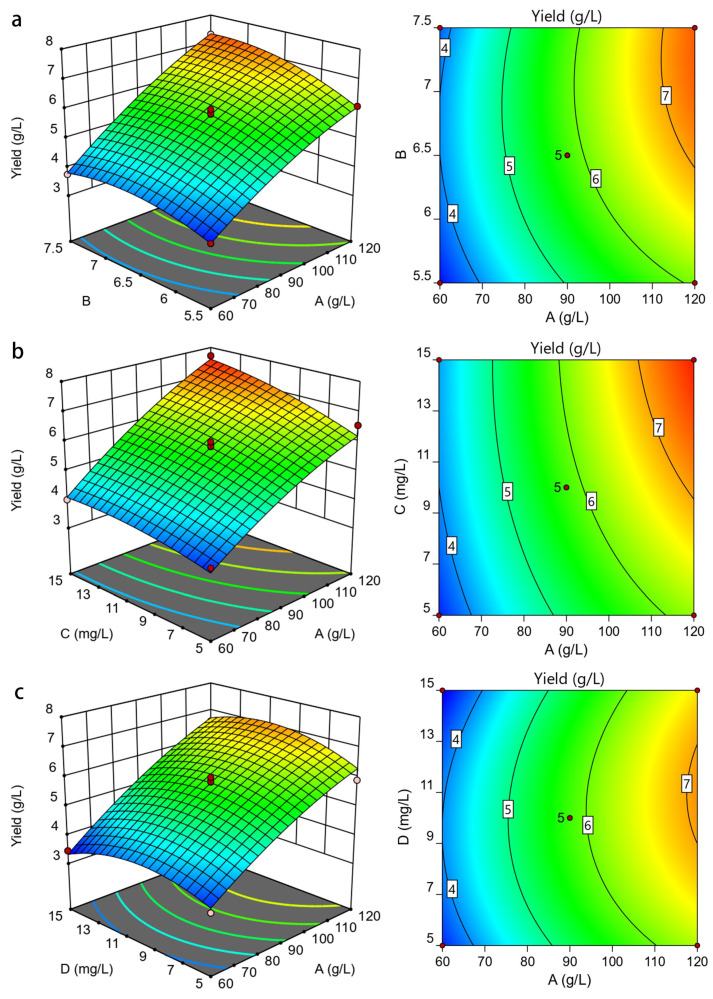
3D response surface and 2D contour plots. ((**a**): glucose concentration and pH; (**b**): glucose concentration and Mn^2+^ concentration; (**c**): glucose concentration and Zn^2+^ concentration; (**d**): pH and Mn^2+^ concentration; (**e**): pH and Zn^2+^ concentration; (**f**): Mn^2+^ concentration and Zn^2+^ concentration).

**Figure 3 foods-11-00999-f003:**
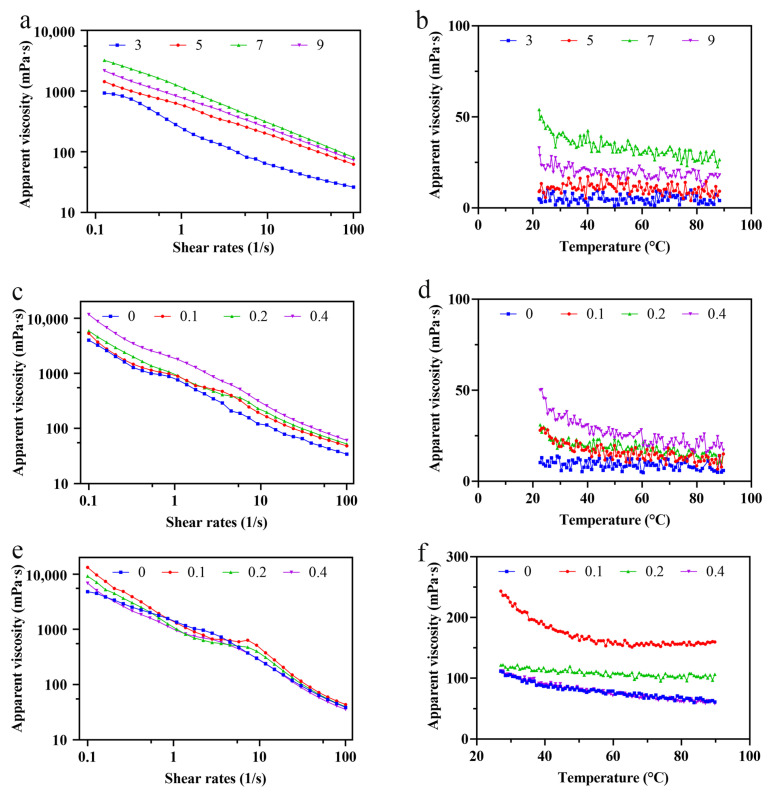
The apparent viscosity dependence of LSEP solutions on shear rates at different pH (**a**), CaCl_2_ concentration (**c**), and NaCl concentration (**e**); the apparent viscosity dependence of LSEP solutions on temperature at different pH (**b**), CaCl_2_ concentration (**d**), and NaCl concentration (**f**).

**Figure 4 foods-11-00999-f004:**
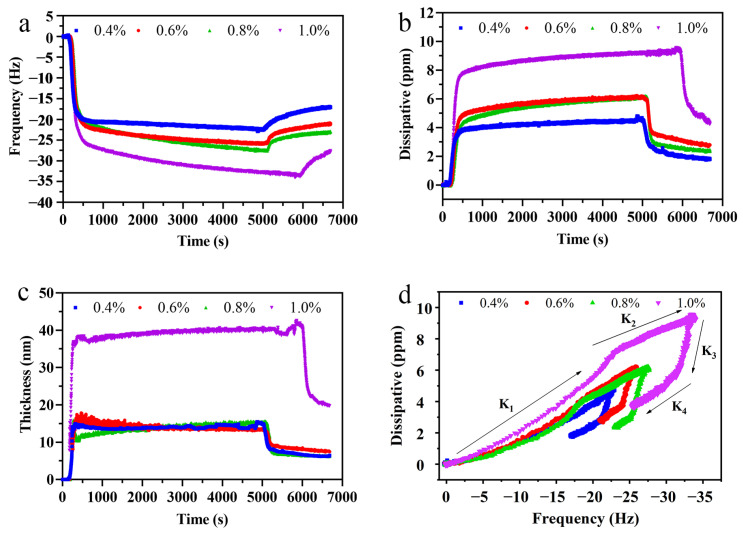
The frequency (**a**), dissipative (**b**) and thickness (**c**) during the adhesion and washing at the MCT-coated sensors; D-f plots (**d**) during adhesion and washing processes (K_1_, K_2_, K_3_, and K_4_ were the slopes at four different phases, respectively).

**Table 1 foods-11-00999-t001:** Process variables and their ranges of BBD.

Level	−1	0	1
A	60	90	120
B	5.5	6.5	7.5
C	5	10	15
D	5	10	15

**Table 2 foods-11-00999-t002:** The Box–Behnken design and corresponding observed values.

Run	A	B	C	D	Yield (g/L)	
	Glucose (g/L)	pH	Mn^2+^ (mg/L)	Zn^2+^ (mg/L)	Experimental	Predicted
1	60	5.5	10	10	3.45 ± 0.03 ^p^	3.45
2	120	5.5	10	10	6.10 ± 0.05 ^d^	6.07
3	60	7.5	10	10	3.75 ± 0.04 ^o^	3.80
4	120	7.5	10	10	7.30 ± 0.06 ^b^	7.32
5	90	6.5	5	5	4.45 ± 0.05 ^l^	4.62
6	90	6.5	15	5	5.90 ± 0.07 ^f^	5.39
7	90	6.5	5	15	4.00 ± 0.02 ^n^	4.53
8	90	6.5	15	15	5.85 ± 0.03 ^f^	5.69
9	60	6.5	10	5	3.40 ± 0.01 ^p^	3.55
10	120	6.5	10	5	5.90 ± 0.06 ^f^	6.29
11	60	6.5	10	15	3.45 ± 0.03 ^p^	3.33
12	120	6.5	10	15	6.60 ± 0.05 ^c^	6.72
13	90	5.5	5	10	4.35 ± 0.05 ^m^	4.38
14	90	7.5	5	10	5.35 ± 0.03 ^h^	5.15
15	90	5.5	15	10	4.85 ± 0.02 ^k^	5.32
16	90	7.5	15	10	5.90 ± 0.06 ^f^	6.14
17	60	6.5	5	10	3.70 ± 0.01 ^o^	3.54
18	120	6.5	5	10	6.55 ± 0.05 ^c^	6.18
19	60	6.5	15	10	4.00 ± 0.03 ^n^	4.08
20	120	6.5	15	10	7.70 ± 0.08 ^a^	7.57
21	90	5.5	10	5	4.45 ± 0.03 ^l^	4.26
22	90	7.5	10	5	5.40 ± 0.05 ^h^	5.38
23	90	5.5	10	15	4.95 ± 0.03 ^j^	4.68
24	90	7.5	10	15	5.25 ± 0.05 ^i^	5.16
25	90	6.5	10	10	5.45 ± 0.02 ^g^	5.81
26	90	6.5	10	10	5.83 ± 0.04 ^f^	5.81
27	90	6.5	10	10	5.90 ± 0.03 ^f^	5.81
28	90	6.5	10	10	5.85 ± 0.06 ^f^	5.81

The values of different tags in the same rank are significant (*p* < 0.05, *n* = 3) by Duncan’s test.

**Table 3 foods-11-00999-t003:** The ANOVA details of the fitted equation model for the production of LSEP.

Source	Sum of Squares	df ^a^	Mean Square	F-Value	*p*-Value	Significance ^b^
Model	36.24	14	2.59	22.79	<0.0001	**
A-Glucose	28.21	1	28.21	248.44	<0.0001	**
B-pH	1.92	1	1.92	16.91	0.0011	**
C-Mn	2.80	1	2.80	24.69	0.0002	**
D-Zn	0.0300	1	0.0300	0.2642	0.6153	
AB	0.2025	1	0.2025	1.78	0.2031	
AC	0.1806	1	0.1806	1.59	0.2279	
AD	0.1056	1	0.1056	0.9301	0.3512	
BC	0.0006	1	0.0006	0.0055	0.9419	
BD	0.1056	1	0.1056	0.9301	0.3512	
CD	0.0400	1	0.0400	0.3522	0.5623	
A^2^	0.4864	1	0.4864	4.28	0.0575	
B^2^	0.9065	1	0.9065	7.98	0.0135	*
C^2^	0.2252	1	0.2252	1.98	0.1809	
D^2^	2.04	1	2.04	18.00	0.0008	**
Residual	1.59	14	0.1136			
Lack of Fit	1.41	10	0.1414	3.22	0.1356	Not significant
Pure Error	0.1757	4	0.0439			
Cor Total	37.83	28				
Source	Sum of Squares	df ^a^	Mean Square	F-value	*p*-value	Significance ^b^

R^2^ = 0.9580; Adjusted R^2^ = 0.9159; C.V. % = 6.45; Adeq Precision = 17.5357. ^a^ df, Degrees of freedom. ^b^ * significant influence (*p* < 0.05); ** highly significant influence (*p* < 0.01).

**Table 4 foods-11-00999-t004:** The slopes of all D-f curves, including K_1_, K_2_, K_3_, and K_4_ stages.

	K_1_ (LR)	K_2_ (LR)	K_3_ (LR)	K_4_ (LR)
0.4%	−0.1242 ± 0.00199 (0.9650) ^a^	−0.2780 ± 0.00088 (0.9711) ^c^	−1.2710 ± 0.02397 (0.9386) ^c^	−0.2389 ± 0.00149 (0.9622) ^a^
0.6%	−0.1959 ± 0.00187 (0.9797) ^c^	−0.2759 ± 0.00068 (0.9837) ^c^	−1.1649 ± 0.01166 (0.9574) ^b^	−0.3420 ± 0.00213 (0.9640) ^c^
0.8%	−0.1851 ± 0.00225 (0.9630) ^b^	−0.2401 ± 0.00045 (0.9901) ^b^	−0.9431 ± 0.02375 (0.8561) ^a^	−0.3819 ± 0.00523 (0.8424) ^d^
1.0%	−0.2852 ± 0.00205 (0.9880) ^d^	−0.1859 ± 0.00044 (0.9803) ^a^	−1.8835 ± 0.05091 (0.9435) ^d^	−0.3298 ± 0.00202 (0.9632) ^b^

The values of different tags in the same rank are significant (*p* < 0.05, *n* = 3) by Duncan’s test. LR represents linear regression coefficients.

## Data Availability

Data is contained within the article.

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
