# Peer review of "Optimization and Rheological Study of an Exopolysaccharide Obtained from Fermented Mature Coconut Water with Lipomyces starkeyi"

_foods, 2022, doi:10.3390/foods11070999_

Round 1

Reviewer 1 Report

The manuscript entitles “Optimization and rheological properties of an exopolysaccharide fermented by Lipomyces starkeyi with mature coconut water” investigated the reheological properties of exopolysaccharide fermented solution.  The subject frame of the work is well constructed. So, in this respect and this article should be contributed to present research. I recommended this work for publication after the following minor revisions.

  1. There are several typographical mistakes as well in whole manuscript. Therefore, the author’s thoroughly careful check the language and typo mistake to minimize the error.
  2. The abstract should be beginning with a sentence about the background of concept and the aims as well as novelty of study should be mentions. What exactly is the novelty of this study? The abstract is poorly written and should be improved. Abbreviations must be avoided in abstract. Parenthesis should be avoided in abstract - this is poor writing. Please improve.
  3. I did not understand, author use crude EPS or purified. Need clarification in materials and methods section. Also improved results and discussion section.
  4. Introduction; Check and format the citations in the whole manuscript. Also, Appropriate references must be provided to explained the background, what is already done and why this study carried out. Other vise the novelty of this research is still poorly presented. This is important especially for the high IF journals. The scientific style should be used. What exactly is the aim of this work? Hypothesis statement is missing in the introduction section.
  5. Results and discussion; General remark to the discussion - In my opinion, the discussion provided by Authors is difficult to follow and verify due missing critical details in the methodology section. Due to poorly described material and poorly presented methods, I am not able to follow and properly review the discussion.
  6. All figures are of poor technical quality and not suitable for publication, especially in a high reputed journal. Font size and kind is too small and must be unified in all figures. Small writings are unreadable. All figures must be self-explanatory. Axis titles are poorly presented or absent. Units are missing. Are the data presented in figures significantly different? At least error bars should be shown.
  7. I suggest first time write full name rather than abbreviation; revise throughout in manuscript

Reviewer 2 Report

The authors tried optimizing the yield and rheological properties of the exopolysaccharide produced by L. starkeyi using coconut water through a response surface methodology.  The submitted manuscript seems to be well written and concisely summarized the result.  Although I think that the manuscript substantially satisfies the criteria for publication, there are some minor concerns should be solved before acceptance in the present form as follows:

  1. I feel uneasy with the title. Could you change the title as: “... exopolysaccharide obtained from fermented mature coconut water with Lipomyces starkeyi” (just recommendation)

  1. The authors seem to aim to apply the extracted or purified L. starkeyi-derived EPS to food additive and excipient. If so, are there any reports on the health-promoting effect and traditional application of the ESP produced by L. starkeyi (or the yeast itself)? The description regarding the potential functions may increase the worth of the EPS and the manuscript.  Additionally, how about adding information of L. starkeyi about history of usage as a food.

  1. Are there any additional advantages on use of coconut water besides waste loss? The water seems nutrient rich, while it contains rich minerals? Additionally, are there any reports on the fermentation of other juices that contains Zn2+ and Mn2+ with L. starkeyi or other microorganisms including probiotics?
